# Phenotypic identification of Metallo-ß-lactamase resistance Gram negative bacteria from a clinical specimen in Sidama, Ethiopia

**Tsegaye Alemayehu**[1]*, **Wondwesson Abera**[1], **Musa Mohammed Ali**[1],
**Bethelihem Jimma**[1], **Henok Ayalew**[2], **Limenih Habte**[3], **Frezer Teka**[4], **Demissie Asegu**[1]

**1** Hawassa University College of Medicine and Health Sciences, Hawassa, Ethiopia, **2** St. Paul's Hospital Millennium Medical College, Addis Ababa, Ethiopia, **3** Medewelabu University College of Medicine and Health Science, Shashemene, Ethiopia, **4** SNNPR Public Health Institute, Hawassa, Ethiopia

* alemayehutsegaye@ymail.com

**Data Availability Statement:** All relevant data are within the paper and its Supporting Information files.

**Funding:** The author(s) received no specific funding for this work.

## Abstract

### Background

Metallo-beta lactamase resistance is one of the carbapenem resistances that worsen the world nowadays. A new variant of carbapenem-resistant has only limited reports from Africa including Ethiopia. This study aimed to determine Metallo -ß- lactamase resistance Gram-negative bacteria in Hawassa University Comprehensive Specialized Hospital January–June 2023.

### Method

A cross-sectional study was conducted in which consecutive patients infected with Gram-negative bacteria were included in the study. A structured questionnaire was used to collect the data with oriented nurses if the patients/or caregivers gave consent to participate in the study. Clinical specimens are processed based on the standard operating procedure of the Microbiology laboratory and Clinical laboratory standard institute guidelines. Culture and sensitivity testing was used to isolate the bacteria. Gram staining and biochemical tests was used to identify the bacteria to genus and species. Kirby disc diffusion technique was used to determine the susceptibility of antibiotics. Statistical Software for Social Science (SPSS) version 21 is used for data entry and analysis. Descriptive statistics and logistic regression were used to interpret the data. The odds ratio at 95% confidence interval (CI) and p-value < 0.05 were taken as a statistically significant association.

### Result

Our study included 153 isolates from different specimens, 83 (54.2%) were from male patients and 70 (45.8%) were from females. *Klebsiella pneumonia* was the predominant 43, followed by *Escherichia coli* 32, *Acinetobacter* spp 25, *Pseudomonas* spp 15, *Enterobacter agglomerus* 9, *Klebsiella ozaenae* 6, *Enterobacter cloacae* 5, *Klebsiella oxytoca* 4, (*Klebsiella rhinoscleromatis*, *Proteus mirabilis* and *Morganella morganii*) 3, *Providencia stuartii* 2 and (*Citrobacter* spp & *Proteus vulgaris*) 1. The rates of multi, extensive and pan-drug

**Competing interests:** The authors have declared that no competing interests exist.

resistance bacteria accounted for 128/153 (83.7%), 77 /153(50.3%), and 26/153 (17.0%), respectively. Carbapenem resistance was 21 (13.7%), of this 7.2% were Enterobacteriaceae, 5.2% were *Acetinobacter* spp. and 1.3% *Pseudomonas* spp. Metallo-beta-lactamase was 17 (11.1%), of this, Enterobacteriaceae were 9(5.9%), *Acetinobacter* spp. 7(4.6%), and *Pseudomonas* spp. 1(0.7%). There were no variables statistically significantly associated with metallo-beta-lactamase-resistant.

## Conclusion

Our study revealed that Metallo-beta-lactamase resistance was circulating in the study area. There was a high rate of carbapenem resistance, multi, extensive and pan-drug resistance. Therefore, a measure should be taken to alleviate the emerging threat that leaves the patients without the option of treatment.

## Introduction

Antimicrobial resistance (AMR) is the most perplexing clinical and public health issue listed as one of the leading causes of death for people around the world. Without having a full understanding of all the contributing causes and without any measures to address them, it is anticipated that by 2050, AMR will be responsible for 10 million annual fatalities [1]. According to the level of relative antibiotic resistance and the urgency of the situation, specialists from the World Health Organisation [2] and Germany classified microorganisms as critical, high, and medium priority in 2017. This places Carbapenem-resistant Enterobacteriaceae (CRE), *Pseudomonas aeruginosa (P. aeruginosa)*, and *Acinetobacter baumannii (A. baumannii)* in the critical priority category [3].

The last-resort therapy for Enterobacteriaceae with multidrug resistance (MDR) is carbapenem, which is also known as imipenem, meropenem, ertapenem and doripenem. The synthesis of acquired Metallo-beta lactamases (MBLs) is primarily blamed for Enterobacteriaceae complex carbapenem resistance (CR) mechanisms of MBLs [4]. The three classifications A, B, and D of the Ambler classification of Extended spectrum beta lactamase (ESBL) include the enzymes that hydrolyze carbapenems. The three main types of MBLs, which are carbapenem class B enzymes, are Imipenemase, Verona integron-encoded Metallo-beta-lactamase (VIM), and New-Delhi Metallo-beta-lactamase (NDM) enzymes [5].

The most significant ESBLs are carbapenemase, notably MBLs since they can hydrolyze all beta-lactams, including carbapenems, except monobactams [6]. Mobile genetic elements that promote horizontal gene transfer (HGT) across bacteria and have a high capacity for spreading typically carry MBL-encoding genes [7, 8]. The most significant nosocomial infections are those that produce MBL, and if they continue to spread throughout healthcare facilities, there will be a serious global concern [9]. As a result, active surveillance is required to identify the frequency and prevalence of MBL-producing bacteria in the neighbourhood and to aid in the containment of their expansion [10].

To the best of our knowledge, there are few publications on the presence of MBLs in clinical isolates [11, 12], however, there have been a few investigations on carbapenem resistance in Ethiopia [11–13]. This investigation seeks to identify the phenotypic composition of non-fermenting gram-negative bacteria and Enterobacteriaceae resistant to metallo-ß-lactamase in Hawassa University Comprehensive Specialised Hospital (HUCSH).

## Materials and methods

### Description of the study area

The study was conducted at HUCSH from January to June 2023. HUCSH is located in the capital city of the Sidama regional state at Hawassa, 275 kilometres (KMs) away from Addis Ababa, the capital city of Ethiopia. The altitude of the town is 1697m above sea level with a mean annual temperature and rainfall of 20.9˚C and 997.6 ml, respectively. Hawassa University Comprehensive Specialized Hospital was established in November 2005 and it serves about 12 million people. Patients seeking medical care receive services at different outpatient and inpatient units (surgery, gynaecology, obstetrics, internal medicine, paediatrics, ophthalmology, psychiatry, radiology, and pathology). The laboratory in the hospital analyses arrays of tests including parasitological, microbiological, immunological, haematological, and biochemical analyses. In the microbiology section, all aerobic culture and sensitivity testing are performed regularly from Monday to Sunday for 24 hours.

### Study subjects

All patients who visited the microbiology laboratory for routine culture and sensitivity testing during the study period were the source population. All patients confirmed with Enterobacteriaceae and non-fermenter Gram-negative bacteria were the study population. All patients who requested culture and susceptibility and those who volunteered to participate in the study were included in the study. Patients who refused to participate in the study and patients with gram-positive isolates were excluded from the study.

### Study design

A prospective cross-sectional study was used to select the clinical isolates that were included in the study. A convenient sampling technique was used to collect 153 Enterobacteriaceae and non-fermenter Gram-negative bacteria in which consecutive patients with gram-negative isolates were enrolled.

### Data collection

Metallo beta-lactamase resistance, extended-spectrum beta-lactamase, carbapenem resistance, multidrug resistance, extensive drug resistance, and pan-drug resistance were the dependent variables. Whereas, sociodemographic factors, age, sex, residence, and other clinical features including malnutrition, presence of chronic disease, previous antibiotics usage, external device usage, and ward type were independent variables. Sociodemographic and clinical data were collected with a structured questionnaire with oriented nurses from the patient's and patients' charts.

### Laboratory diagnosis

All the samples were collected routinely for culture and susceptibility testing using the standard operating procedure (SOP) of the microbiology laboratory for each specimen. The samples were inoculated on appropriate culture media based on the essentiality of the samples. Blood culture was collected with the sterile procedure and immediately inoculated to Tryptone soy broth (TSB) at the site of collection. Then the bottle was transported for incubation in the microbiology laboratory. It was incubated at 37˚C for five days [14]. The bottle was checked daily for the presence/or absence of growth indicators i.e., gas, pellicles, clot, and haemolysis. The sample was sub-cultured on blood agar plate (BAP), Chocolate agar plate (CAP), and MacConkey agar (MAC), and Gram stain was conducted at 24 hours of incubation even if

there were no growth indicators. Identification was performed based on their gram staining characteristics. Gram-negative bacteria were included in this study. Finally, on day five, the bottle was sub-cultured on solid media and was reported as having no growth. Urine and puss were inoculated with BAP and MAC. Chocolate agar was included for Ear discharge and Nasal swab. The stool specimen was cultured on MAC and xylose lysine deoxycholate agar (XLD).

## Biochemical testing

Once the organism was identified as Gram-negative in Gram staining, serial biochemical testing that was prepared routinely was performed to identify the isolates. Triple sugar iron agar, urea, citrate, mannitol fermentation, lysine iron agar, sulphur indole motility testing, and oxidase were used to identify the isolates to species level.

## Antibiotics susceptibility testing

The Kirby disk diffusion technique was used to perform susceptibility testing on Muller Hinton agar (MHA) [15]. Twelve different antibacterial were used including Augmentin—AUG (10 µg) (cotrimoxazole—COT (1.25/23.75µg), ceftazidime- CAZ (30µg), ceftriaxone—CTR (30µg), imipenem- IMP (10µg), gentamycin- GEN (10µg), chloramphenicol—CAF (30µg), ampicillin—AMP (10µg), ciprofloxacin—CIP (5µg), cefotaxime—CTX (30µg) nitrofurantoin —NIT (5 µg), and piperacillin-tazobactam—PIT (100/10 µg). It was interpreted based on the clinical laboratory standard institute (CLSI) guideline as sensitive, intermediate, and resistant by measuring the zone of inhibition. Carbapenem resistance and Metallo beta-lactamase resistance were determined using a CLSI [16].

## Carbapenem inactivation method (CIM) for confirmation of carbapenemase

A 1 mL loop of Enterobacteriaceae or a 10 mL loop of *P. aeruginosa* from Blood agar plates was emulsified in 2 mL trypticase soy broth (TSB) and added to the modified carbapenem inactivation method (mCIM). The suspension was then applied to an imipenem disc, which was subsequently incubated for 4 hours at 37˚C. Using the direct colony suspension technique, an *Escherichia coli* American type culture collection (*E. coli* ATCC) 25922 0.5 McFarland suspension was created in saline. Using the standard disc diffusion technique, *E. coli* ATCC 25922 was infected into a Mueller Hinton agar (MHA) plate. The *E. coli* ATCC 25922 indicator strains were previously used to inoculate an MHA plate before the imipenem disc was taken out of the TSB. For 18–24 hours, plates were incubated at 35˚C in free air. Colonies within a 16–18 mm zone or an inhibition zone diameter of 6–15 mm were regarded as favorable results, whereas an inhibition zone diameter of 19 mm was regarded as a negative result [16].

## Simplified carbapenem inactivation method (sCIM)

The mCIM is the foundation of the sCIM, which has improved experimental methods. In the sCIM, the organism to be examined was smeared immediately onto an antimicrobial disc rather than being incubated for 4 hours in the organism culture media as in the mCIM. Following the normal disc diffusion approach, a 0.5 McFarland standard suspension (using the direct colony suspension method) of *E. coli* ATCC 25922 was injected onto the MHA plate to perform the sCIM for *Acinetobacter* species. For 3 to 10 minutes, plates were left to dry. Then, 1–3 overnight colonies of the test organisms developed on blood agar were spread over one side of an imipenem disc (10µg), and shortly after, the MHA plate that had previously been injected with *E. coli* ATCC 25922 was placed on the side of the disc with bacteria. As a control,

an imipenem disc was positioned on an MHA plate. All plates were incubated in room temperature air for 16–18 hours at 35˚C. The susceptible indicator strain expanded unrestrained because bacteria that generated Carbapenemase can hydrolyze imipenem. The zone of inhibition around the disc has a diameter of 6–20 mm, and satellite growth of *E. coli* ATCC 25922 colonies around the disc with a zone diameter of 22 mm was considered carbapenemase positive; a zone of inhibition 26 mm was considered a negative result; a zone of inhibition of 23–25 mm was considered a carbapenemase indeterminate result [17].

## Identification of Metallo-β-lactamase resistance

The discovered Enterobacteriaceae, *P. aeruginosa*, and sCIM-positive *Acinetobacter* species underwent additional MBL screening because they are known to be inhibited by EDTA-based metal chelators. A Metallo carbapenemase producer was defined as a 5-mm increase in zone diameter for EDTA- modified carbapenem inactivation method (eCIM) compared to zone diameter for mCIM. Metallo-carbapenemase were deemed to be negative if there was a zone diameter difference of 4 mm between the eCIM and mCIM [18].

## Data management and analysis

Data were coded, entered, and processed with SPSS statistical software version 21 and were presented by table and graph. The bivariate and multivariate logistic regression model was used to check the predictors of a dependent variable. The odds ratio at 95% confidence level and p-value $< 0.05$ was considered statistical significance.

## Ethics approval and consent to participate

Hawassa University College of Medicine and Health Sciences institutional review board (IRB) (Ref. No: IRB/068/14 and Date: 01/10/2022) has approved the proposal. At that moment support letter was gained from the hospital management. Data were collected after written informed consent and/or agreement gained from each child's parents and patients. All information is kept secret using codes and locking on the board. The result of the patient-reported to the clinician within three or four days and those who were culturally positive were treated accordingly.

## Quality control

Data quality was ensured using standardized data collection materials, pretesting of the questionnaires, proper orientation of data collectors before the start of data collection, and intensive supervision during data collection by the authors. For laboratory analysis, pre-analytical, analytical, and post-analytical stages of quality assurance incorporated in the standard operating procedures (SOPs) of the microbiology laboratory were strictly followed. Besides, a well-trained and experienced microbiologist was participating in the laboratory analysis procedure. Media sterility was checked after preparation and incubating for 24 hours. Quality control strains such as *E. coli* (ATCC-25922), and *P. aeruginosa* (ATCC-27853) were obtained from the Ethiopian Public Health Institute (EPHI) to check the characteristics of the colony while growing respective media and biochemical tests.

# Result

## Sociodemographic characteristics

Our study included 153 patients with different infections confirmed having Gram negative bacteria. Of these, eighty-three 83 (54.2%) were from male patients, and seventy 70 (45.8%)

were from females. Most of the patients that is 98 were from under 10 years which is 61.4%, whereas 34 (22.2%) and 21 (13.7%) were from (11–20) years and > 31 years respectively. The mean and standard deviation of the age of the patients was 12.3 ± 17.2 years, which ranged (from 1 day to 70 years). Ninety-eight (64.1%) of the patients were from urban residences (**Table 4**).

## Clinical features of the study subjects

Most of the study subjects were inpatients 125 (81.7%), from the pediatric ward, 50 (32.7%), and a fourth of the study subjects did not have previous hospital stays. Regarding the length of hospital stay, 34 (22.2%) stayed for (6–10 days), most infections were from community-acquired infections (CAI) 81 (52.9%), and bloodstream infections were the most common infections 68 (44.4%). Most patients used external devices 124 (81%), from this most of them used more than two types of external devices that is 62 (40.5%). Almost three-fourths of the patients used antibiotics before culture and sensitivity were done and have taken three or more antibiotics classes 57 (37.3%) (Table 4).

## Frequency of isolates

Of 153 isolates, *Klebsiella pneumonia* (*K. pneumonia)* was the predominant isolated 43, followed by *E. coli* 32, *Acinetobacter* spp 25, *Pseudomonas* spp. 15, *Enterobacter agglomerus* (*E. agglomerus) 9, Klebsiella ozaenae* (*K. ozaenae)* 6, *Enterobacter cloacae* (*E. cloacae) 5, Klebsiella oxytoca* (*K. oxytoca) 4, [Klebsiella rhinoscleromatis* (*K. rhinoscleromatis), Proteus mirabilis* (*P. mirabilis) and Morganella morganii* (*M. morganii)] 3, Providencia stuartii* (*P. stuartii) 2 and* [*Citrobacter spp & Proteus vulgaris* (*P. vulgaris)] 1* (Fig 1).

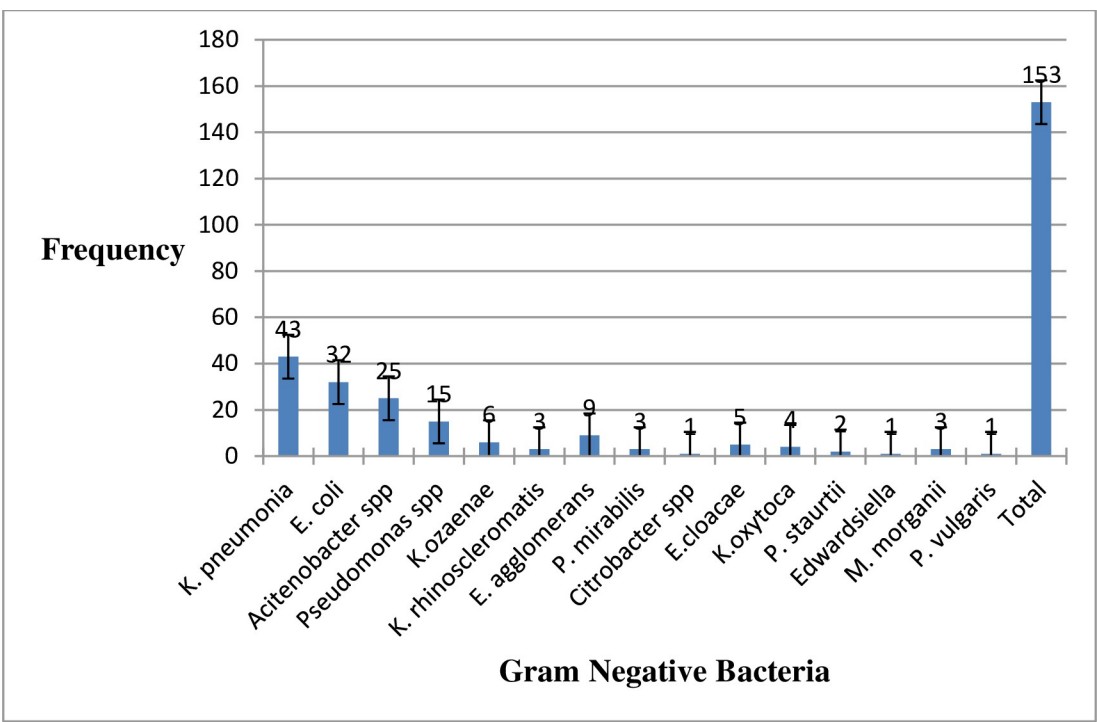

**Fig 1. The frequency of bacterial isolates from clinical specimens in HUCSH, 2023.**

**Table 1. Antimicrobial resistance patterns of Gram-negative isolates at HUCSH, 20230.**

| | Antibiotics | | | | | | | | | | | |
|---|---|---|---|---|---|---|---|---|---|---|---|---|
| | CAZ (153) | CTX (13) | CTR (139) | AMP (57) | AUG (112) | CN (153) | CIP (153) | NIT (42) | IMP (153) | COT (121) | PT (150) | CAF (83) |
| | R (%) | R (%) | R (%) | R (%) | R (%) | R (%) | R (%) | R (%) | R (%) | R (%) | R (%) | R (%) |
| *K. pneumonia* | 40 (93) | 42 (97.7) | 40 (93.0) | NR | 35 (81.4) | 32 (74.4) | 22 (51.2) | 4 (9.3) | 8 (18.6) | 36 (83.7) | 15 (34.9) | 15 (34.9) |
| *E. coli* | 24 (75) | 27 (84.4) | 27 (84.4) | 31 (96.9) | 24 (75.0) | 12 (37.5) | 21 (65.6) | 11 (34.4) | 2 (6.2) | 28 (87.5) | 11 (34.4) | 6 (18.8) |
| *Acinetobacter* spp. | 22 (88) | 23 (92) | 23 (92.0) | NR | NR | 19 (76.0) | 17 (68.0) | NR | 17 (68) | 8 (32) | 13 (52.0) | NR |
| *Pseudomonas* spp. | 10 (66.7) | NR | NR | NR | NR | 6 (40.0) | 6 (40.0) | NR | 6 (40.0) | NR | 6 (40.0) | NR |
| *K. ozaenae* | 6 (100) | 6 (100) | 6 (100) | NR | 4 (66.7) | 5 (83.3) | 4 (66.7) | 3 (50.0) | 1 (16.7) | 6 (100) | 2 (33.3) | 0 (0.0) |
| *K. oxytoca* | 3 (75) | 4 (100) | 4 (100) | NR | 2 (66.7) | 2 (50.0) | 3 (75.0) | 1 (25.0) | 0 (0.0) | 4 (100) | 1 (25.0) | 2 (50.0) |
| *K. rhinoscleromatis* | 3 (100) | 3 (100) | 3 (100) | NR | 3 (100) | 3 (100) | 2 (66.7) | 1 (33.3) | 1 (33.3) | 3 (100) | 1 (33.3) | 2 (66.7) |
| *E. agglomerus* | 1 (11.1) | 1 (11.1) | 3 (33.3) | 8 (88.9) | 1 (11.1) | 1 (11.1) | 1 (11.1) | 0 (0.0) | 0 (0.0) | 1 (11.1) | 1 (11.1) | 0 (0.0) |
| *E. cloacae* | 4 (80.0) | 4 (80) | 5 (100) | 5 (100) | 5 (100) | 2 (40.0) | 2 (40.0) | 1 (20.0) | 0 (0.0) | 5 (100) | 1 (20.0) | 2 (40.0) |
| *P. mirabilis* | 3 (100) | 3 (100) | 3 (100) | 3 (100) | 3 (100) | 1 (33.3) | 2 (66.7) | 1 (33.3) | 1 (33.3) | 3 (100) | 1 (33.3) | 1 (33.3) |
| *M. morganii* | 3 (100) | 2 (66.7) | 3 (100) | 3 (100) | 2 (66.7) | 1 (33.3) | 2 (66.7) | 3 (100) | 2 (66.7) | 2 (66.7) | 2 (66.7) | 3 (100) |
| *P. stuartii* | 1 (50) | 1 (50) | 1 (50) | 2 (100) | 1 (50.0) | 1 (50.0) | 1 (50.0) | 0 (0.0) | 0 (0.0) | 1 (50) | 1 (50) | 1 (50.0) |
| *Citrobacter* spp. | 0 (0) | 0 (0) | 0 (0.0) | 1 (100) | 1 (100) | 0 (0.0) | 0 (0.0) | 0 (0.0) | 0 (0.0) | 0 (0.0) | 0 (0.0) | 0 (0.0) |
| *P. vulgaris* | 1 (100) | 1 (100) | 1 (100) | 1 (100) | 1 (100) | 0 (0.0) | 1 (100) | 0 (0.0) | 0 (0.0) | 1 (100) | 1 (100) | 1 (100) |
| *Edwardsiella* | 0 (0) | 0 (0) | 0 (0.0) | 0 (0) | 0 (0) | 0 (0.0) | 0 (0.0) | 0 (0.0) | 0 (0.0 | 0 (0.0) | 0 (0.0) | 0 (0.0) |
| Total | 121 (79.1) | 121 (84.6) | 119 (85.6) | 69 (94.5) | 82 (73.2) | 85 (55.6) | 84 (54.9) | 25 (59.5) | 38 (24.8) | 98 (80.9) | 56 (37.3) | 29 (37.7) |

**Keynote**: Ceftazidime- CAZ, cefotaxime–CTX, ceftriaxone—CTR, ampicillin—AMP, Cotrimoxazole- COT, imipenem- IMP, gentamycin- GEN, chloramphenicol–CAF, ciprofloxacin—CIP, nitrofurantoin–NIT, and piperacillin-tazobactam–PIT.

## Antimicrobial resistance

A high resistance pattern was observed for ampicillin 69 (94.5%), followed by ceftriaxone 119 (85.6%), cefoxitin 121 (84.6%), cotrimoxazole 98 (80.9%), ceftazidime 121 (79.1%), augmentin 82 (73.2%), nitrofurantoin 25 (59.5%), gentamicin 85 (55.6%), ciprofloxacin 84 (54.9%), chloramphenicol 29 (37.7%), piperacillin-tazobactam 56 (37.3%) and meropenem 38 (24.8%) (Table 1).

## MDR, XDR and PDR

Our study outlined that MDR, XDR, and PDR bacteria accounted for 128 (83.7%), 77 (50.3%), and 26 (17.0%) respectively. ESBL-suspected bacteria accounted for 119 (77.8%) (Table 2).

## Carbapenem and Metallo-beta lactamase resistance

Our study indicates that 38 (24.8%) isolates were suspected as carbapenem-resistant with the Imipenem disk test, and from this 22 (14.4, 95% CI: 8.7–20.0%) isolates were confirmed as carbapenemase producers [CRE: 7.2 (3.3–11.1%), *Acetobacter* spp. 5.2 (2.0–9.2%) and *Pseudomonas* spp. 1.3 (0.0–3.3%)]. Of this, 17 [11.1 (6.5–16.3%)] were confirmed as metallo-beta-lactamase classes with eCIM [Enterobacteriaceae 9 [5.9 (2.6–9.8%)], *Acetinobacter spp.* 7 [4.6 (2.0–8.5%)], and *Pseudomonas* spp. 1 [(0.7 (0.0–2.0%)] (Table 3).

## Associated factors

In this study, statistical analysis was conducted to check for the risk factors for MBL resistance, however, there were no variables associated with these infections with p-value < 0.05 (Table 4).

**Table 2. The prevalence of MDR, XDR, PDR, and ESBL among bacteria from clinical specimens, HUCSH, 2023.**

| Isolates | MDR | | XDR | | PDR | | ESBL-S | |
|---|---|---|---|---|---|---|---|---|
| | Yes (%) | No (%) | Yes (%) | No (%) | Yes (%) | No (%) | Yes (%) | No (%) |
| *K. pneumonia* | 41 (95.3) | 2 (4.7) | 22 (51.2) | 21 (48.8) | 1 (2.3) | 42 (97.7) | 39 (90.7) | 4 (9.3) |
| *E. coli* | 28 (87.5) | 4 (12.5) | 17 (53.1) | 15 (46.9) | 1 (3.1) | 31 (96.9) | 25 (78.1) | 7 (21.9) |
| *Acinetobacter* spp | 22 (88.0) | 3 (12.0) | 19 (76.0) | 6 (24.0) | 17 (68.0) | 8 (32.0) | 22 (88.0) | 3 (12.0) |
| *Pseudomonas* spp | 11 (73.3) | 4 (26.7) | 5 (33.3) | 10 (66.7) | 4 (26.7) | 11 (73.3) | 10 (66.7) | 5 (33.3) |
| *K. ozaenae* | 5 (83.3) | 1 (16.7) | 3 (50.0) | 3 (50.0) | 1 (16.7) | 5 (83.3) | 4 (66.7) | 2 (33.3) |
| *K. rhinoscleromatis* | 3 (100) | 0 (0.0) | 3 (100.0) | 0 (0.0) | 1 (33.3) | 2 (66.7) | 2 (66.7) | 1 (33.3) |
| *K. oxytoca* | 4 (100) | 0 (0.0) | 3 (75.0) | 1 (25.0) | 0 (0.0) | 4 (100) | 3 (75.0) | 1 (25.0) |
| *E. agglomerus* | 1 (11.1) | 8 (88.9) | 0 (0.0) | 9 (100 | 0 (0.0) | 7 (100) | 1 (11.1) | 8 (88.9) |
| *E. cloacae* | 5 (100) | 0 (0.0) | 0 (0.0) | 5 (100) | 0 (0.0) | 5 (100) | 5 (100) | 0 (0.0) |
| *P. mirabilis* | 3 (100) | 0 (0.0) | 1 (33.3) | 2 (66.7) | 1 (33.3) | 2 (66.7) | 3 (100) | 0 (0.0) |
| *Citrobacter* spp | 0 (0.0) | 1 (100) | 0 (0.0) | 1 (100) | 0 (0.0) | 1 (100) | 0 (0.0) | 1 (100) |
| *P. stuartii* | 1 (50) | 1 (50) | 1 (50.0) | 1 (50.0) | 0 (0.0) | 2 (100) | 1 (50.0) | 1 (50.0) |
| *Edwardsiella* | 0 (0.0) | 1 (100) | 0 (0.0) | 1 (100) | 0 (0.0) | 1 (100) | 0 (0.0) | 1 (1)00 |
| *M. morganii* | 3 (100) | 0 (0.0) | 2 (66.7) | 1 (33.3) | 0 (0.0) | 3 (100) | 3 (100) | 0 (0.0) |
| *P. vulgaris* | 1 (100) | 0 (0.0) | 1 (100.0) | 0 (0.0) | 0 (0.0) | 1 (100) | 1 (100) | 0 (0.0) |
| Total | 128 (83.7) | 25 (16.3) | 77 (50.3) | 76 (49.7) | 26 (17.0) | 127 (83.0) | 119 (77.8) | 34 (22.2) |

## Discussion

Most metallo-beta- lactamases quickly hydrolyze, especially the carbapenems, and they are resistant to metallo-lactamase inhibitors. These enzymes were first identified as chromosomal enzymes in a selected few species, but they are now also discovered in several plasmids that are distributed throughout the majority of the planet [19].

Our study signposts that 14.0 (8.7–20.0%) isolates were confirmed as carbapenemase producers, this is comparable with the study from India 17.0% [20], 16.8% [21], 15.63% [22],

**Table 3. Carbapenem and metallo-beta-lactamase resistance among bacteria isolated from clinical specimens at HUCSH, 2023.**

| Isolates | CR-S | | CR-C | | MBL | |
|---|---|---|---|---|---|---|
| | Yes (%) | No (%) | Yes (%) | No (%) | Yes (%) | No (%) |
| *K. pneumonia* | 8 (18.6) | 35 (81.4) | 6 (14.0) | 37 (86.0) | 5 (11.6) | 38 (88.4) |
| *E. coli* | 2 (6.2) | 30 (93.8) | 1 (3.1) | 31 (96.9) | 1 (3.1) | 31 (96.9) |
| *Acinetobacter* spp | 17 (68.0) | 8 (32.0) | 8 (34.8) | 15 (65.2) | 7 (28.0) | 18 (72.0) |
| *Pseudomonas* spp | 6 (40.0) | 9 (60.0) | 2 (13.3) | 13 (86.7) | 1 (6.7) | 14 (93.3) |
| *K. ozaenae* | 1 (16.7) | 5 (83.3) | 1 (16.7) | 5 (83.3) | 1 (16.7) | 5 (83.3) |
| *K. rhinoscleromatis* | 1 (33.3) | 2 (66.7) | 1 (33.3) | 2 (66.7) | 1 (33.3) | 2 (66.7) |
| *K. oxytoca* | 0 (0.0) | 4 (100) | 0 (0.0) | 4 (100) | 0 (0.0) | 4 (100) |
| *E. agglomerus* | 0 (0.0) | 9 (100) | 0 (0.0) | 9 (100) | 0 (0.0) | 9 (100) |
| *E. cloacae* | 0 (0.0) | 4 (100) | 0 (0.0) | 5 (100) | 0 (0.0) | 5 (100) |
| *P. mirabilis* | 1 (33.3) | 2 (66.7) | 1 (33.3) | 2 (66.7) | 1 (33.3) | 2 (66.7) |
| *Citrobacter* spp | 0 (0.0) | 1 (100) | 0 (0.0) | 1 (100) | 0 (0.0) | 1 (100) |
| *P. stuartii* | 0 (0.0) | 2 (100) | 0 (0.0) | 2 (100) | 0 (0.0) | 2 (100) |
| *Edwardsiella* | 0 (0.0) | 1 (100) | 0 (0.0) | 1 (100) | 0 (0.0) | 1 (100) |
| *M. morganii* | 2 (66.1) | 1 (33.3) | 2 (66.7) | 1 (33.3) | 0 (0.0) | 1 (100) |
| *P. vulgaris* | 0 (0.0) | 1 (100) | 0 (0.0) | 1 (100) | 0 (0.0) | 1 (100) |
| *Total* | **38 (24.8)** | **115 (75.2)** | **22 (14.4)** | **129 (84.3)** | **17 (11.1)** | **136 (88.9)** |

**Table 4. Assessment of associated factors among bacteria isolated from clinical specimens at HUCSH, 2023.**

| Variables | Frequency (%) | MBL | | COR (95% CI) | P-value |
|---|---|---|---|---|---|
| | | Yes (%) | No (%) | | |
| **Age (years)** | | | | | |
| <10 | 98 (64.1) | 11 (11.2) | 87 (88.8) | .537 (.153–1.89) | .333 |
| 11–20 | 34 (22.2) | 2 (5.9) | 32 (94.1) | .27 (.044–1.601) | .148 |
| >31 | 21 (13.7) | 4 (19.0) | 17 (81.0) | 1 | |
| **Sex** | | | | | |
| Male | 83 (54.2) | 8 (9.6) | 75 (90.4) | 1 | |
| Female | 70 (45.8) | 9 (12.9) | 61 (87.1) | 1.38 (.50–3.800) | .529 |
| **Residence** | | | | | |
| Urban | 98 (64.1) | 11 (11.2) | 87 (88.8) | 1.03 (.36–2.96) | .953 |
| Rural | 55 (35.9) | 6 (10.9) | 49 (89.1) | 1 | |
| **Patient Status** | | | | | |
| Inpatient | 125 (81.7) | 13 (10.4) | 112 (89.6) | .69 (.20–2.32) | .556 |
| Outpatient | 28 (18.3) | 4 (14.3) | 24 (85.7) | 1 | |
| **Ward** | | | | | |
| NICU | 43 (28.1) | 5 (11.6) | 38 (88.4) | 5.26 (.59–47.14) | .138 |
| Pedi | 50 (32.7) | 5 (10.0) | 45 (90.0) | 4.44 (.50–39.67) | .182 |
| Medical | 19 (12.4) | 6 (31.6) | 13 (68.4) | 18.46 (2.0–167.9) | .010 |
| Others | 41 (26.8) | 1 (2.4) | 40 (97.6) | 1 | |
| **Previous Hospital Stay** | | | | | |
| Yes | 42 (27.5) | 5 (11.9) | 37 (88.1) | 1.12 (.37–3.38) | .848 |
| No | 111 (72.5) | 12 (10.8) | 99 (89.2) | 1 | |
| **Hospital Stay (in days)** | | | | | |
| <1 | 33 (21.6) | 4 (12.1) | 29 (87.9) | 1.01 (.21–4.99) | .989 |
| 2–5 | 28 (18.3) | 2 (7.1) | 26 (92.9) | .56 (.09–3.69) | .550 |
| 6–10 | 34 (22.2) | 3 (8.8) | 31 (91.2) | .71 (.13–3.85) | .691 |
| 11–20 | 33 (21.6) | 5 (15.2) | 28 (84.8) | 1.31 (.28–6.09) | .731 |
| >21 | 25 (16.3) | 3 (12.0) | 22 (88.0) | 1 | |
| **Types of Infection** | | | | | |
| HAI | 72 (47.1) | 9 (12.5) | 63 (87.5) | 1.30 (.48–3.58) | .607 |
| CAI | 81 (52.9) | 8 (9.9) | 73 (90.1) | 1 | |
| **Sites of Infection** | | | | | |
| BSI | 68 (44.4) | 73 (90.1) | 60 (88.2) | .61 (.13–10.76) | .871 |
| UTI | 41 (26.8) | 3 (7.3) | 38 (92.7) | .71 (.07–7.65) | .778 |
| SSI | 19 (12.4) | 2 (10.5) | 17 (89.5) | 1.06 (.08–13.3) | .965 |
| RTI | 15 (9.8) | 3 (20.0) | 12 (80.0) | 2.25 (.20–25.37) | .512 |
| Other | 10 (6.5) | 1 (10.0) | 9 (90.0) | 1 | |
| **External Device** | | | | | |
| Yes | 124 (81.0) | 12 (9.7) | 112 (90.3) | 1.94 (.63–6.04) | .250 |
| No | 29 (19.0) | 5 (17.2) | 24 (82.8) | 1 | |
| **Types of External Devices** | | | | | |
| Single | 15 (9.8) | 1 (6.7) | 14 (93.3) | 1.67 (.18–15.5) | .654 |
| Two | 47 (30.7) | 5 (10.6) | 42 (89.4) | 1.50 (.17–13.49) | .718 |
| More than 2 | 62 (40.5) | 6 (9.7) | 56 (90.3) | 2.92 (.31–27.56) | .350 |
| No | 29 (19.0) | 5 (17.2) | 24 (82.8) | 1 | |
| **Antibiotics Usage** | | | | | |
| Yes | 111 (72.5) | 12 (10.8) | 99 (89.2) | 1.12 (.37–3.38) | .848 |

*(Continued)*

**Table 4.** (Continued)

| Variables | Frequency (%) | MBL | | COR (95% CI) | P-value |
|---|---|---|---|---|---|
| | | Yes (%) | No (%) | | |
| No | 42 (27.5) | 5 (11.9) | 37 (88.1) | 1 | |
| **Class of antibiotics** | | | | | |
| Single | 21 (13.7) | 3 (14.3) | 18 (85.7) | 1.23 (.27–5.74) | .789 |
| Two | 33 (21.6) | 2 (6.1) | 31 (93.9) | .48 (.09–2.63) | .396 |
| Multiple | 57 (37.3) | 7 (12.3) | 50 (87.7) | 1.04 (.31–3.52) | .955 |
| No | 42 (27.5) | 5 (11.9) | 37 (88.1) | 1 | |
| **Underline Disease** | | | | | |
| Yes | 104 (68) | 14 (13.5) | 90 (86.5) | 2.39 (.65–8.72) | .189 |
| No | 49 (32) | 3 (6.1) | 46 (93.9) | 1 | |
| **Malnutrition** | | | | | |
| Yes | 65 (42.5) | 7 (10.8) | 58 (89.2) | 1 | |
| No | 88 (57.5) | 10 (11.4) | 78 (88.6) | 1.06 (.38–2.96) | .908 |

11.96% [23]) and Hawassa 9% [13]. In contrast to our finding, a high rate of CR was reported from Sudan 83 % [24], Egypt 70% [25], Thailand 69.7% [26], Saudi Arabia 58.23% [27], Egypt 54.1% [28], Egypt 29% [29], and lower rates reported from Ghana 5.7% [26]. This difference might be due to antibiotic usage style, the presence of good infection control, availability of surveillance and reporting, the difference of genetic factors of bacteria and the presence of effective of public health interventions.

Our study identified the prevalence of carbapenem resistance Enterobacteriaceae was 7.2 (3.3–11.1%), this finding is comparable with the findings from Nigeria at 8% [30], 7.9% [31], 6.5% [32], a surveillance study in Africa 6.2% [33], Poland 6% [34], Addis Ababa 5.4% [12] and Hawassa 4.5% [13]. In contrast to our result an increased resistance rate was reported from Uganda 22.4% [35], Kuwait 14.0% [30], and a lower than reported from Vietnam 0.6% [36].

In this study, the rate of carbapenem resistance *Acinetobacter* spp was 5.2 (2.0–9.2%). This is in line with a study from, Switzerland 8.4% [37], Germany 4.4% [38] and Indonesia 3.47% [39]. However, it is lower than studies from, Makkah and Al-Madinah at 82.5% [40], Oman at 80.4% [41], a systematic review and meta-analysis in Africa at 56.97% [42], Uganda at 31% [43], Addis Ababa 61% [44], New Caledonia 24.8% [45], Lebanon 22.8% [46], in sub-Saharan African countries 20% [47], Poland 16% [34]. It is higher than reported from Hawassa at 1.8% [13], a systematic review and meta-analysis of Pakistan at 0.28% [48].

Our study also identified the prevalence of carbapenem resistance *Pseudomonas* spp. was 1.3 (0.0–3.3%) agreeing with the study from Hawassa 2.7% [13]. However, a higher finding was reported in Brazil 43.9% [49], China 41.3% [50], Saudi Arabia 37.2% [51], Oman 29.9% [41], Cameroon 25.1% [52], Africa 21.4% [42], Lebanon 24.8% [46], Uganda 24% [43], Addis Ababa 22% [44], Algeria 18.75% [53], China 18.4% [54], Poland at 8% [34], in sub-Saharan African countries 8% [47] and Africa 4.5% [33]. The variation in CRE prevalence observed in our study compared to other regions can be attributed to a complex interplay of factors including antibiotic usage patterns, infection control measures, surveillance robustness, microbial ecology, public health interventions, and socioeconomic factors. Understanding these differences is crucial for designing targeted strategies to combat antibiotic resistance, highlighting the need for tailored interventions based on regional specificities and resource availability.

In this finding, metallo-beta-lactamase resistance Gram negative bacteria was 11.1 (6.5–16.3%). In contrast to our finding, a higher rate of Metallo-beta lactamase was reported from

Egypt at 50% [25], India 44.1% [55], Pakistan 36.7% [56] and Sudan 36.1% [49]. However, a lower rate was reported from India 2.9% [57] and a systematic review and meta-analysis Pakistan 0.34% [48].

In this study, metallo-beta lactamase Enterobacteriaceae was 5.9 (2.6–9.8%), lower than a study reported from India 1.25% [57]. In our study, metallo-beta lactamase resistance *Acetinobacter spp.* was 4.6 (2.0–8.5%), which is comparable with studies from Tehran Iran 3.48% [58]. In contrast to our study, a higher result was reported from Iran at 93.3% [59] and Morocco at 42.5% [60].

In this study, Metallo beta lactamase resistance *Pseudomonas* spp. was 0.7 (0.0–2.0%), a higher result reported in a study from China at 55.2% [61], Iran 40% [59], Pakistan 33.9% [56], a systematic review and meta-analysis in Egypt 33.7% [62], Brazil 30.4% [63], India 20.0% [64], India 8.05% [65] and India 4.7% [66]. The observed variation in MBL resistance rates highlights the complex interplay of factors influencing antibiotic resistance. Our findings underscore the importance of understanding regional contexts when interpreting resistance data and developing strategies to combat antibiotic resistance. By comparing our results with those from different regions, we emphasize the need for tailored interventions that address local healthcare practices, antibiotic use, and surveillance capabilities to effectively manage and reduce the burden of antibiotic resistance.

Our study outlined that multidrug resistance bacteria accounted for 83.7 (77.8–89.5%) of Gram-negative bacteria. This is in line with a study from a systematic review and meta-analysis of 82.7% [67]. In contrast to our finding higher rates of multidrug resistance were reported from a studies Nepal at 91.3% [68]. However, a lower rate was reported from a systematic review and meta-analysis in Ethiopia 70.56% [69], India 66.12% [70], Saudi Arabia 64.3% [71], Kuwait 38.7% [72], India 37.1% [73], a systematic review and meta-analysis data from USA 27% [74] and Germany 4.31% [75].

Similarly, our finding signposts that extensive drug resistance Gram-negative bacteria was 50.3 (41.8–58.8%), this is in line with a study from Bangladesh at 51% [76] and higher than a study from, India at 34.32% [70], Ethiopia 32.2% [77], India 13.8% [73], Saudi Arabia 12.1% [78]. However, it is lower than a study from Bangladesh 72% [79], Egypt 65% [80], and Pakistan 64% [81].

Our study revealed that pan drug resistance bacteria accounted for 17.0 (11.8–24.2%), which is higher than a study from Ethiopia (8.95% [82], 7.3% [77]), India (0.98% [70], 0% [73]) and Saudi Arabia 0% [78]. This difference might be due to antibiotic usage that can be mis use or over use, the presence of effective antibiotic stewardship, the implementation and adherence to infection control measures in hospital, differences in the robustness and coverage of surveillance systems, the availability and quality of diagnostic facilities, clonal spread resistance gene, differences in national healthcare policies. Regions with limited resources may struggle to implement effective control measures, economic disparities can affect healthcare quality and access, impacting the prevalence of resistant infections., and travel and migration status of the country also one of the possibilities for disparities of drug resistance.

## Conclusion

Our study indicated that there was a high rate of carbapenem and metallo beta lactamase resistance in the isolated Gram-negative bacteria. Besides this, the rates of MDR, XDR, and PDR were high. *K. pneumonia* was the predominant isolate and *Citrobacter* spp and *P. vulgaris* were the list isolates. Ampicillin resisted more and Imipenem was less resistant. This shows MBL is circulating in the hospital. Therefore, a strict infection prevention and control mechanism should be taken to reduce the transmission. Beside this, an advanced technique should be employed to characterize the MBL in the region with one health approach.

## Limitations of the study

Due to the lack of some antibiotics in the local market, our study cannot determine the confirmation of ESBL, and due to the lack of budget for molecular testing carbapenemase were confirmed only phenotypically using culture and sensitivity testing method, which is recommended with CLSI.

## Supporting information

**S1 Dataset.**
(XLSX)

## Acknowledgments

First, we would like to thank Hawassa University for giving us this opportunity. Second, we would like to thank the staff of HUCSH Microbiology Laboratory for their support during the analysis. Finally, we would give our heartfelt thanks to the study participants.

## Author Contributions

**Conceptualization:** Tsegaye Alemayehu, Demissie Asegu.

**Data curation:** Tsegaye Alemayehu, Musa Mohammed Ali, Henok Ayalew.

**Formal analysis:** Tsegaye Alemayehu, Musa Mohammed Ali, Frezer Teka, Demissie Asegu.

**Funding acquisition:** Tsegaye Alemayehu, Wondwesson Abera, Musa Mohammed Ali, Bethelihem Jimma, Frezer Teka, Demissie Asegu.

**Investigation:** Tsegaye Alemayehu, Wondwesson Abera, Musa Mohammed Ali, Demissie Asegu.

**Methodology:** Tsegaye Alemayehu, Wondwesson Abera, Musa Mohammed Ali.

**Project administration:** Tsegaye Alemayehu.

**Resources:** Tsegaye Alemayehu, Bethelihem Jimma, Henok Ayalew, Limenih Habte, Frezer Teka.

**Software:** Tsegaye Alemayehu.

**Supervision:** Tsegaye Alemayehu, Musa Mohammed Ali, Limenih Habte, Demissie Asegu.

**Validation:** Tsegaye Alemayehu, Bethelihem Jimma, Henok Ayalew, Demissie Asegu.

**Visualization:** Tsegaye Alemayehu, Henok Ayalew.

**Writing – original draft:** Tsegaye Alemayehu, Demissie Asegu.

**Writing – review & editing:** Tsegaye Alemayehu, Musa Mohammed Ali, Demissie Asegu.

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
