## [Decision Letter · Decision Letter 0]

5 Jun 2024

PONE-D-24-02706Phenotypic Identification of Metallo-ß- lactamase resistance Gram Negative bacteria from a clinical specimen in Sidama, EthiopiaPLOS ONE

Dear Dr. Alemeyhu,

Thank you for submitting your manuscript to PLOS ONE. After careful consideration, we feel that it has merit but does not fully meet PLOS ONE’s publication criteria as it currently stands. Therefore, we invite you to submit a revised version of the manuscript that addresses the points raised during the review process.

We look forward to receiving your revised manuscript.

Kind regards,

Mihret Tilahun, MSc

Academic Editor

PLOS ONE

Journal Requirements:

2. During our evaluation of the documents provided, we noted that your ethics approval letter did not cover the entire date range for participant recruitment. Before we can proceed further with the submission, please provide the ethics approval extension document for the study. If the document is in another language, please also provide an English translation. Please note that if this document is not included when your manuscript is resubmitted, it may be rejected 

3. Ethics statement appears in the Methods section of the manuscript AND at the end of the manuscript:

Your ethics statement should only appear in the Methods section of your manuscript. If your ethics statement is written in any section besides the Methods, please delete it from any other section. 

4. Please amend the manuscript submission data (via Edit Submission) to include author Henok Ayalew.

Additional Editor Comments:

To the author please revise your paper based on the the reviwer comments and the language needs revision

please attach authore response point by point and the clear and highlighted revised manuscript

Reviewers' comments:

Reviewer's Responses to Questions

**Comments to the Author**

1. Is the manuscript technically sound, and do the data support the conclusions?

Reviewer #1: Partly

Reviewer #2: Yes

2. Has the statistical analysis been performed appropriately and rigorously? 

Reviewer #1: Yes

Reviewer #2: Yes

3. Have the authors made all data underlying the findings in their manuscript fully available?

Reviewer #1: No

Reviewer #2: Yes

4. Is the manuscript presented in an intelligible fashion and written in standard English?

Reviewer #1: Yes

Reviewer #2: Yes

5. Review Comments to the Author

Reviewer #1: The manuscript titled “Phenotypic Identification of Metallo-ß- lactamase resistance Gram Negative bacteria from a clinical specimen in Sidama, Ethiopia” is important in that the authors have highlighted the detection of Carbapenemase and MBL producers by phenotypes of Gram-negative bacteria in Sidama, Ethiopia. Undoubtedly the study has a profound impact as the WHO (2017) highlighted drug-resistant GNB as a priority pathogen.

It can be understood that there is no laboratory molecular setup in developing countries to perform confirmatory tests. As well as low-budget research it is difficult to do molecular tests. However, the study has several deficits, and gaps without addressing these concerns the manuscript is not recommended for publication.

The authors have shown only Carbapenem resistance without testing tigecycline and polymyxin-b how XDR and PDR strain detection which is unclear in the methods section.

It is important to learn whether these stains have colistin or other last-resort antibacterial agents’ resistance.

There are several recent papers the authors have missed, there is no comparison or discussion between these papers and the author’s findings.

The discussion should be improved and the conclusion should be revised.

It makes no sense merely comparing percentage occurrence of one trait or phenotype with other studies. The methods used to detect these traits may be different for the previous studies. Just comparing if the percent occurrence in the present study was similar to, or greater or smaller than the previous study does not suggest anything in the current frame of discussion. The authors need to elaborate more on the results obtained in the present study and draw inferences from the same.

Research results should be highlighted as the major outcome and recommendation is required.

Reviewer #2: In Page1:

City and country of each author should be added to the Authors’ affiliation.

Abstract:

• Page 1: Method section needs more explanation about the practical work.

• Page 2, lines 40-42: The sentence “The rates of multi, extensive and pan-drug resistance bacteria accounted for 83.7%, 50.3%, and 17.0%, respectively” needs more explanation. What do these numbers mean?

Introduction

• Page 3, Line 67: What does the abbreviation “ESBLs” mean?

• Page 3, Line 89: Correct “20.9 o C”.

Materials and Methods

• Page 4, line 101: delete “&”.

• Page 4, line 110: delete “Study Methodology”

• Page 4, line 125: correct “37 C0”

• Page 5, line 142: correct the sentence “Twelve different antibacterial will be performed” to be “Twelve different antibacterial agents were used”.

• Page 5, line 142: Write the full name of “mCIM” and similar abbreviations in other pages.

• Page 6, lines 186, 187: the sentence “Gram-negative bacilli that produced serine carbapenemases were those that were carbapenemases positive but Metallo carbapenemases negative” is not necessary.

• Page 6, line 195: add the IRB number.

Results:

• Some typing errors need correction in the whole results section, specifically spaces between numbers.

• Page 7, line 225: what do you mean by this sentence “from this most of them 226 used more than two types of external devices 62 (40.5%)”?

• Page 8: abbreviations of the antibiotics used should be added to the Table 1 capture.

• Page 10, line 259: correct the word “statical” to be “statistical”.

Discussion:

• Please remove some “comma symbol where is not necessary.

• Please revise the text without “Italic” typing other than the name of bacteria.

• Write the full name for the abbreviations.

• Some typing errors should be revised.

Conclusion:

• Page 13, line 323: correct the word “Impenem” to “imipenem”

6. PLOS authors have the option to publish the peer review history of their article (what does this mean?). If published, this will include your full peer review and any attached files.

Reviewer #1: No

Reviewer #2: No

---

## [Author Response · Author response to Decision Letter 0]

21 Jul 2024

Point by point response

General Response

Dear Editor and reviewers, thank you very much for your expertise and time to review and edit our manuscript. As per your recommendation we modify and include the data set as supporting file. Here we tried to make a modification based on your comments. Even if, the reviewer 1 comments are not specific and difficult to address as per your comments we tried to address a general response. For reviewer 2 since his comment is specific, we tried to address point by point and shaded on the manuscript with yellow color for both modifications.

Reviewer 1:

Comment: The manuscript titled “Phenotypic Identification of Metallo-ß- lactamase resistance Gram Negative bacteria from a clinical specimen in Sidama, Ethiopia” is important in that the authors have highlighted the detection of Carbapenemase and MBL producers by phenotypes of Gram-negative bacteria in Sidama, Ethiopia. Undoubtedly the study has a profound impact as the WHO (2017) highlighted drug-resistant GNB as a priority pathogen.

Response: Thank for recognizing our work

Comment: It can be understood that there is no laboratory molecular setup in developing countries to perform confirmatory tests. As well as low-budget research it is difficult to do molecular tests. However, the study has several deficits, and gaps without addressing these concerns the manuscript is not recommended for publication.

The authors have shown only Carbapenem resistance without testing tigecycline and polymyxin-b how EDR and PDR strain detection which is unclear in the methods section. It is important to learn whether these stains have colistin or other last-resort antibacterial agents’ resistance. 

Response: We write this idea in the limitation but the CLSI recommend as to check the presence of Carbapenem resistance phenotypically with carbapenem class of antibiotics. At this time, we couldn’t improve as per your comment. I hope you understand the shortage of antibiotics in the local market.

Comment: There are several recent papers the authors have missed, there is no comparison or discussion between these papers and the author’s findings.

The discussion should be improved and the conclusion should be revised.

It makes no sense merely comparing percentage occurrence of one trait or phenotype with other studies. The methods used to detect these traits may be different for the previous studies. Just comparing if the percent occurrence in the present study was similar to, or greater or smaller than the previous study does not suggest anything in the current frame of discussion. The authors need to elaborate more on the results obtained in the present study and draw inferences from the same.

Response: We tried to modify the discussion as per your comment and add a justification.

Comment: Research results should be highlighted as the major outcome and recommendation is required.

Response: We tried to modify the conclusion and recommendation.

Reviewer #2: 

Comment:

In Page 1:

City and country of each author should be added to the Authors’ affiliation.

Response: modified as per your comment

Comment: 

Abstract:

• Page 1: Method section needs more explanation about the practical work.

Response: modified 

• Page 2, lines 40-42: The sentence “The rates of multi, extensive and pan-drug resistance bacteria accounted for 83.7%, 50.3%, and 17.0%, respectively” needs more explanation. What do these numbers mean?

Response: modified

 Comments:

Introduction

• Page 3, Line 67: What does the abbreviation “ESBLs” mean?

Response: modified

• Page 3, Line 89: Correct “20.9 o C”.

Response: modified

Comments: 

Materials and Methods

• Page 4, line 101: delete “”.

Response: modified

• Page 4, line 110: delete “Study Methodology”

Response: Accepted

• Page 4, line 125: correct “37 C0”

Response: modified

• Page 5, line 142: correct the sentence “Twelve different antibacterial will be performed” to be “Twelve different antibacterial agents were used”

Response: modified

.• Page 5, line 142: Write the full name of “mCIM” and similar abbreviations in other pages.

Response: modified

• Page 6, lines 186, 187: the sentence “Gram-negative bacilli that produced serine carbapenemase were those that were carbapenemase positive but Metallo carbapenemase negative” is not necessary.

Response: modified

• Page 6, line 195: add the IRB number.

Response: modified

Comments

Results:

• Some typing errors need correction in the whole results section, specifically spaces between numbers.

Response: modified

• Page 7, line 225: what do you mean by this sentence “from this most of them 226 used more than two types of external devices 62 (40.5%)”?

Response: this is modified

• Page 8: abbreviations of the antibiotics used should be added to the Table 1 capture.

Response: modified

• Page 10, line 259: correct the word “statical” to be “statistical”.

Response: modified

Comments

Discussion:

• Please remove some “comma symbol where is not necessary.

Response: modified

• Please revise the text without “Italic” typing other than the name of bacteria.

Response: modified

• Write the full name for the abbreviations.

Response: modified

•Some typing errors should be revised.

Response: modified

Comments

Conclusion:

• Page 13, line 323: correct the word “Impenem” to “imipenem”

Response: modified

The end

---

## [Editor Report · Decision Letter 1]

22 Aug 2024

PONE-D-24-02706R1Phenotypic Identification of Metallo-ß- lactamase resistance Gram Negative bacteria from a clinical specimen in Sidama, EthiopiaPLOS ONE

Dear Dr. Alemeyhu,

Thank you for submitting your manuscript to PLOS ONE. After careful consideration, we feel that it has merit but does not fully meet PLOS ONE’s publication criteria as it currently stands. Therefore, we invite you to submit a revised version of the manuscript that addresses the points raised during the review process.

During our evaluation of the documents provided, we noted that your ethics approval letter did not cover the entire date range for participant recruitment and data collection. Before we can proceed further with the submission, please provide the ethics approval extension document(s) for the study that cover the study period from 1st March 2023 - June 2023. If the document is in another language, please also provide an English translation. Please note that if this document is not included when your manuscript is resubmitted, it may be rejected.

We look forward to receiving your revised manuscript.

Kind regards,

Emma Campbell, Ph.D

Staff Editor

PLOS ONE

Journal Requirements:

2. During our evaluation of the documents provided, we noted that your ethics approval letter did not cover the entire date range for participant recruitment and data collection. Before we can proceed further with the submission, please provide the ethics approval extension document(s) for the study that cover the study period from 1st March 2023 - June 2023. If the document is in another language, please also provide an English translation. Please note that if this document is not included when your manuscript is resubmitted, it may be rejected.

3. Please amend the manuscript submission data (via Edit Submission) to include author Henok Ayalew.

---

## [Author Response · Author response to Decision Letter 1]

3 Sep 2024

Dear Editor, 

Thank you very much for your effort to make our manuscript imporved a lot through regress review. Here we made a modification as per your comments and suggestion and attached as point by point response.

---

## [Editor Report · Decision Letter 2]

24 Oct 2024

Phenotypic Identification of Metallo-ß- lactamase resistance Gram Negative bacteria from a clinical specimen in Sidama, Ethiopia

PONE-D-24-02706R2

Dear Dr. Alemeyhu,

We’re pleased to inform you that your manuscript has been judged scientifically suitable for publication and will be formally accepted for publication once it meets all outstanding technical requirements.

Kind regards,

Mihret Tilahun, MSc

Academic Editor

PLOS ONE
---

## [Editor Report · Acceptance letter]

4 Nov 2024

PONE-D-24-02706R2 

PLOS ONE

Dear Dr. Alemayehu, 

I'm pleased to inform you that your manuscript has been deemed suitable for publication in PLOS ONE. Congratulations! Your manuscript is now being handed over to our production team.

Kind regards, 

on behalf of

Dr. Mihret Tilahun 

Academic Editor

PLOS ONE